# Study on Oxygen Evolution Reaction of Ir Nanodendrites Supported on Antimony Tin Oxide

**DOI:** 10.3390/nano13152264

**Published:** 2023-08-07

**Authors:** Yu-Chun Chiang, Zhi-Hui Pu, Ziyi Wang

**Affiliations:** 1Department of Mechanical Engineering, Yuan Ze University, Taoyuan 320, Taiwan; s1080821@mail.yzu.edu.tw (Z.-H.P.); s1080857@mail.yzu.edu.tw (Z.W.); 2Fuel Cell Center, Yuan Ze University, Taoyuan 320, Taiwan

**Keywords:** iridium nanodendrites, antimony tin oxide, oxygen evolution reaction

## Abstract

In this study, the iridium nanodendrites (Ir NDs) and antimony tin oxide (ATO)-supported Ir NDs (Ir ND/ATO) were prepared by a surfactant-mediated method to investigate the effect of ATO support and evaluate the electrocatalytic activity for the oxygen evolution reaction (OER). The nano-branched Ir ND structures were successfully prepared alone or supported on ATO. The Ir NDs exhibited major diffraction peaks of the fcc Ir metal, though the Ir NDs consisted of metallic Ir as well as Ir oxides. Among the Ir ND samples, Ir ND2 showed the highest mass-based OER catalytic activity (116 mA/mg at 1.8 V), while it suffered from high degradation in activity after a long-term test. On the other hand, Ir ND2/ATO had OER activity of 798 mA/mg, and this activity remained >99% after 100 cycles of LSV and the charge transfer resistance increased by less than 3 ohm. The enhanced durability of the OER mass activities of Ir ND2/ATO catalysts over Ir NDs and Ir black could be attributed to the small crystallite size of Ir and the increase in the ratio of Ir (III) to Ir (IV), improving the interactions between the Ir NDs and the ATO support.

## 1. Introduction

Hydrogen is a clean and efficient energy carrier. When hydrogen is used in the replacement of fossil fuels, the immediate benefits include a reduction in emissions. Water electrolysis is well known as a practical way to produce hydrogen with high purity, which is green when the power is from renewable energy sources [1,2]. Compared to the conventional alkaline electrolysis, the proton exchange membrane water electrolysis (PEMWE) is an alternative and promising technique because of several advantages including high efficiency at high current densities, ease of maintenance, system compactness, and rapid response to startups and shutdowns [2,3,4]. However, the application of PEMWE is limited by the high cost and high loading of the anodic electrocatalysts. The high cost is associated with the use of high loadings of noble metals, where the high loadings are due to the sluggish kinetics of the oxygen evolution reaction (OER) [5]. Noble metals, such as iridium (Ir) and ruthenium (Ru) as well as their oxides (IrO_2_ and RuO_2_), are currently available electrocatalysts for the OER to provide high corrosion resistance and good catalytic activity [6,7,8,9,10,11]. Since noble metals are expensive and scarce, lowering the loading of the OER noble metal catalyst is necessary for the mass production of hydrogen based on this technology.

The most proposed reaction pathway for OER in acidic media is that of two H_2_O molecules being oxidized into one oxygen molecule by releasing four H^+^ ions and four electrons [11]. Since four electrons are transferred, the mechanisms comprise multistep adsorption–desorption reaction processes. The dissolution processes during the cycles with and without hydrous oxide buildup on an iridium metal electrode were suggested [12], and the mechanism of iridium dissolution triggered by OER was also proposed by Cherevko et al. [13]. Ru-based materials have high OER activity; however, they suffer from rapid dissolution due to the formation of volatile tetroxide (RuO_4_) in OER potential regions; thus, their durability issues must be addressed to ensure practical use [14,15]. In contrast, Ir-based materials possess compatible electrochemical activity and durability and are now the most widely used electrocatalysts for the OER in acid media [16,17].

An increase in the utilization of catalysts facilitates a decrease in the use of catalysts. One attractive strategy for addressing this challenge is to use a support material, which must possess a high surface area, a porous and interconnected structure, corrosion resistance, and high electrical conductivity [18]. The use of a high-surface-area support enables an increase in accessible surface area by decreasing the Ir particle size and increasing the surface-to-mass ratio of the catalyst [19,20]. Supported nanocatalysts were observed to be extremely active toward the OER because they possessed mixed Ir oxidation states and a high density of active sites [21]. Carbon materials are used as the support of the electrocatalysts in the applications of the proton exchange membrane fuel cell. However, carbon materials could not resist the high working potential and harshly acidic environment of water electrolysis. The resultant outcome is carbon corrosion, which results in the migration and agglomeration of the nanocatalysts and even detachment from the surface of the support. Therefore, several metal oxides have been studied as alternatives for carbon materials, such as TiO_2_, SnO_2_, doped metal oxides [10,22,23,24], etc.

Most of the metal oxides are semiconductors; thus, doping with hypovalent or hypervalent ions is a prerequisite to high electrical conductivity, but the change in the doping concentration causes severe ohmic losses [18]. Among them, antimony tin oxide (ATO) has attracted much attention because of its nanometer-size scales and relatively high electrical conductivity, compared with most metal oxides. The dispersion of Ir nanoparticles on ATO has been reported to improve the OER activity [10], where the uniform dispersion of Ir nanoparticles and large strong metal–support interaction (SMSI) led to high OER activity of Ir/ATO. It was observed that the introduction of vanadium into ATO support did not provide any more active sites but could change the porosity of the aerogel support and decrease the impurity content of the chlorine [22].

One IrO_2_/ATO catalyst possessing a current density of 63 A/g Ir at an overpotential of 300 mV versus a reversible hydrogen electrode was reported [23], significantly exceeding a commercial TiO_2_-supported IrO_2_ catalyst under the same measurement conditions. Abbou et al. [18] found that the durability of electrochemical activity of IrOx/doped SnO_2_ aerogels was controlled by the resistance to corrosion of the doping element, and by its concentration in the host SnO_2_ matrix. Specifically, Sb-doped SnO_2_ supports continuously dissolve while Ta-doped or Nb-doped SnO_2_ supports with appropriate doping concentrations are stable under acidic OER conditions. This implied that a complex equilibrium relationship existed between SnO_2_ and the doping element oxide.

Another approach is to tailor the structure and morphology of the catalysts. For instance, a nanoneedle network of iridium-containing oxides assembled into macroporous micro-scaled particles [9], nanoporous Ir nanosheets [25], a porous nano net cage of RuIrOx hollow particles [26,27], and amorphous Ir-atomic-cluster-deposited ultrathin IrO_2_ nanoneedles [28] all had high catalytic activity for OER in comparison to a commercial IrO_2_ nanoparticle catalyst. Claudel et al. [29] investigated the degradation mechanisms of OER electrocatalysts and showed that Ir (III) and Ir (V) were the best-performing Ir valences for the OER.

The highly branched structure of the Ir nanodendrites (Ir NDs) with a particle size of ~10 nm provides an increased active facet area that is available for OER in comparison to a commercial Ir catalyst, resulting in enhanced activity toward OER, though the formation of an anodic IrO_2_ film on the surface of the Ir NDs was observed [30]. Oh et al. [5] found that Ir NDs supported on ATO were efficient and stable catalysts for water-splitting reactions. The Ir ND catalysts exhibited a kinetic water-splitting activity twofold larger than supported Ir nanoparticles and eightfold larger than commercial Ir blacks. The size of the nanodendrites was highly related to the reaction temperature; specifically, at a higher temperature, smaller particle sizes were produced by the burst-nucleation process [31]. The formation mechanism of Ir NDs was observed to be a time-dependent process, which suggested that Ir ions were first reduced to Ir^0^ by the reducing agent. Subsequently, the size of the initially formed NPs started to increase as the reaction proceeded. Finally, the NPs self-aggregated, reducing the surface energy, and began to form highly branched dendritic nanostructures through an oriented-attachment formation mechanism [32,33]. Several Ir-based alloy NDs were also studied, including IrPt [34], IrCo [35], and IrW [36], and they usually behaved as bifunctional electrocatalysts.

In this study, the Ir NDs and Ir ND supported on antimony tin oxide (Ir ND/ATO) were prepared. Several material properties were characterized, and the electrocatalytic activity toward OER was evaluated. In addition, the long-term tests and resistance analysis were also investigated. According to the results of this study, ATO can be a good candidate as a support material, and Ir ND/ATO has been demonstrated to be an efficient and stable OER catalyst under a suitable concentration of Ir precursor.

## 2. Materials and Methods

### 2.1. Synthesis of Ir NDs and Ir ND/ATO

The Ir ND nanostructures were prepared by a surfactant-mediated method [5], where three Ir concentrations in the precursor solution were investigated. Firstly, 0.1 mmol dihydrogen hexachloroiridate (IV) hydrate (H2IrCl6·xH2O, 99%, Alfa Aesar) and 10 mmol tetradecyltrimethyl ammonium bromide (TTAB, 99%, Sigma) were dissolved in 75 mL deionized water, where iridium salt: TTAB = 1: 100 mol%, heating to 70 °C with vigorous stirring for 30 min to dissolve the Ir precursor. Then, 50 mg of sodium hydroxide (NaOH, 97%, Riedel-deHaën) was added to the mixture under vigorous stirring, and the color of the solution changed to clear blue. Next, 25 mL ice-cold sodium borohydride (NaBH_4_, 98%, Aldrich) aqueous solution (150 mM) was added dropwise to the mixture under stirring. After that, mixing continued for 6 h at 70 °C, and the solution was cooled down to room temperature naturally. The products were collected by centrifugation and washed several times with ethanol–water mixtures (1:1 *v/v*). Then, the sample was dried at 80 °C in a vacuum oven and denoted as Ir ND1.

In order to determine the effect of the concentration of Ir precursor, in this study, the concentration of Ir precursor was 2- and 5-fold that of the Ir ND1, and all chemicals were also proportionally increased except for the amount of deionized water. The products of the Ir ND prepared with 2- and 5-fold concentrations were named Ir ND2 and Ir ND3. For the preparation of Ir ND/ATO, one commercial ATO (nanopowder, <50 nm, 47 m^2^/g, Aldrich) was dispersed in deionized water (25 mL). This solution was added to the above-mentioned Ir-containing mixture with NaOH, and mixing continued for another 1 h. The following processes were the same as in the previous description.

### 2.2. Characterization of the Samples

Transmission electron microscopy (TEM) was employed to observe the morphology of the Ir NDs nanoparticles and Ir NDs deposited on the ATO support using a transmission electron microscope (Hitachi H-7100, Tokyo, Japan). The electrical conductivity (σ, S/cm) of the ATO was calculated using the equation: σ = 1/(R_sh_ t) [37]. The results were obtained based on the average of ten conductivity measurements on a tablet of 13 mm in diameter. The sheet resistance (R_sh_, Ω) of the tablet was measured using a four-point probe (source meter, Keithley 2401, Radiotek, New Taipei City, Taiwan). The thickness of the tablets (t, cm) was determined using a precise vernier caliper, where the resolution of the scale is 0.01 mm. The elemental compositions of Ir NDs and ATO-supported Ir NDs were acquired by energy dispersive X-ray spectroscopy (EDX), conducted with a QUANTAX Annular XFlash^®^ QUAD FQ5060 (Bruker, Kanagawa, Japan). X-ray diffraction (XRD) was carried out using a powder diffractometer (Rigaku TTRAX III, Tokyo, Japan) equipped with Cu-Kα radiation (λ = 0.15418 nm) to determine the crystal structure. The XRD patterns were collected at a rate of 4°/min over 10–90° in 2θ mode. X-ray photoelectron spectroscopy (XPS) analysis was carried out using a spectrophotometer (PHI 5000 VersaProbe II, ULVAC-PHI, Kanagawa, Japan). The spectrophotometer was equipped with an Al- Kα monochromatic source. The XPSPEAK software (version 4.1) was utilized for the deconvolution of the XPS spectra, which is a nonlinear least-squares curve-fitting program.

### 2.3. Electrochemical Activity Tests

Electrochemical characterization of the electrocatalysts was carried out using a CHI 6116E electrochemical analyzer (CH Instruments, Austin, Texas, USA). A common three-electrode setup and an electrolytic bath filled with 0.1 M HClO_4_ aqueous solution were used to conduct the electrochemical experiments. A glassy carbon rotating disk electrode (RDE, Pine Research Instrument) of 0.5 cm in diameter with a thin layer of the catalyst ink was used as the working electrode. A saturated calomel electrode (SCE) and a platinum wire were used as the reference and counter electrodes, respectively. The catalyst inks, which consisted of 5 mg catalyst, 0.02 mL Nafion^®^ ionomer (5 wt%, DuPont), 2.49 mL deionized water, and 2.49 mL isopropanol [5], were prepared by magnetic stirring for 48 h. Prior to the measurement, the working electrode was activated using a potential sweep between 0.5 and 1.5 V for 100 cycles at a scan rate of 0.1 V/s (a conditioning step). Then, the polarization curves of OER were measured using the linear sweep voltammetry (LSV) for 100 cycles at a scan rate of 0.005 V/s and 1600 rpm at 30 °C in N_2_-saturated 0.1 M HClO_4_ aqueous solution. Moreover, electrochemical impedance spectroscopy (EIS) was performed to determine the ohmic resistance. The EIS spectra of the samples were recorded at 1.8 V with an ac potential amplitude of 5 mA and a frequency range from 1 Hz to 10^5^ Hz.

## 3. Results and Discussion

### 3.1. Physical and Chemical Properties of the Electrocatalysts

TEM images of the samples are shown in Figure 1. It can be seen that Ir NDs could be prepared by the self-assembly of tiny metal seeds using TTAB as an organic capping agent [5,30]. The strong reductant, NaBH_4_, led to the rapid formation of tiny Ir seeds, and the Ir seeds were self-assembled into a dendritic structure through the guidance of TTAB [38]. Figure 1a shows a single Ir-ND1 composed of three-to-five branches in various directions. When the concentration of Ir precursor increased, the number or width of branches increased (Figure 1b,c). The size of each Ir ND was in the order of 5–15 nm. Figure 1d–e confirm that the Ir NDs could be dispersed uniformly on the surface of ATO supports through the in situ synthesis. In addition, the TEM image of one commercial Ir black (Figure 1f) was provided for comparison.

The elemental compositions of the samples were investigated by EDX analysis, as shown in Table 1. For Ir NDs, the Ir content ranged from 88.1 to 95.5 wt.%, similar to that of Ir black (Premetek). For the ATO-supported Ir NDs, the Ir contents in Ir ND1/ATO and Ir ND2/ATO were 20.9 and 26.1 wt.%, respectively, close to the designed percent (20 wt.%). The Sb/(Sn + Sb) was 15% in ATO, while the ratios reduced when Ir NDs were incorporated onto the surface of ATO. It was expected that some Sb ions would be dissolved during the Ir ND deposition processes. The EDX spectrum of Ir ND2/ATO was used as an example, as shown in Figure 2a. The corresponding atomic percentages are also shown in Table 1.

Figure 2b displays the XRD patterns of all samples. Four major peaks on the patterns of Ir NDs at 40.6, 47.3, 69.1, and 83.4° were assigned to diffractions from the (111), (200), (220) and (311) plans of the Ir metallic face-centered cubic (fcc) structure (JCPDS no. 87-0715), respectively. The XRD patterns of the branched Ir NDs suggested that the obtained nanostructures were primarily composed of pure Ir. For Ir ND2/ATO, the major diffraction peaks of metallic Ir were insignificant, indicating the formation of IrOx species on ATO. The crystallite size (d) of Ir nanoparticles was calculated from the Ir (111) peak according to Scherrer’s equation [39], as shown below:(1)$$d=\frac{K\;\lambda }{\beta \cos\theta }$$
where *K* is the Scherrer’s constant of 0.9 [40], *λ* is the X-ray wavelength of Cu-Kα, *β* is the full width at half maximum (FWHM) in radians, and θ is the Bragg angle in radians. The crystallite sizes of Ir ND1, Ir ND2, Ir ND3, and Ir black (Premetek) were 2.28, 1.68, 2.86, and 3.42 nm, respectively.

The chemical states of the samples were investigated by XPS. Figure 3 displays the results of the curve fitting of high-resolution XPS Ir 4f peaks, which confirms the presence of metallic Ir (Ir^0^), Ir (IV), and Ir (III) phases. The latter two may correspond to the molecular formulas Ir(OH)_4_ and Ir(OH)_3_ or IrOx [10]. The existence of Ir oxides may be due to the oxidation of the surface when exposed to air. Table 2 shows the fitting results of the XPS Ir 4f regions for the samples, in which the atomic ratios of Ir 4f were also displayed in the table. Three chemical states of Ir for all samples are illustrated in Figure 4, indicating that the atomic ratio of Ir^0^ increased as the concentration of Ir precursor increased for unsupported Ir NDs, and Ir ND2 comprised the highest percent of Ir (IV). The phase distribution of Ir ND1 was similar to that of Ir (Premetek).

For Ir ND/ATO samples, the percent of Ir^0^ increased in comparison to unsupported Ir NDs, indicating that the deposition of Ir NDs on the surface of ATO contributed to the reduction in Ir precursor. In addition, Ir (III)/Ir (IV) was higher in Ir ND2/ATO than in Ir ND1/ATO, for which it was expected that Ir ND2/ATO might have better electrocatalytic activity [24,29].

Figure 5 shows the curve fitting of high-resolution XPS Sn 3d spectra and the atomic ratios are outlined in Table 3. The as-received ATO contained a small portion of Sn (II), though the majority was still Sn (IV). As seen from Figure 6, the XPS Sn 3d regions were only Sn (IV) for Ir ND/ATO, indicating that the oxidation of Sn (II) to form Sn(IV) promoted the reduction in Ir precursor.

The overlapping of Sb 3d_5/2_ and O 1s regions renders the peak decomposition more complicated. Taking the area ratio of the Sb 3d_3/2_ peak to Sb 3d_5/2_ peak as 2 to 3, the peak decomposition could be accomplished [41], as shown in Figure 6 and Table 4. The oxidation state of Sb on ATO was Sb (V), corresponding to the molecular formula of Sb_2_O_5_/SnO_2_. The deposition of Ir NDs was expected to lead to the dissolution of Sb ions in which the degree of loss was dependent on the Ir content. This implies that Sb was not stable in the alkaline environment during the synthesis process of Ir ND/ATO. However, the Sb amount on Ir ND1/ATO and Ir ND2/ATO remained 3 and 2.03 at.%, respectively. Consequently, the oxidation of Sn and Sb in ATO support suppressed the oxidation of Ir^0^ and Ir (III).

### 3.2. Electrochemical Properties of the Electrocatalysts

The electrocatalytic activity for the OER of the catalysts was evaluated in acidic electrolytes using the LSV process. Figure 7a shows OER polarization curves of three unsupported Ir NDs, two Ir ND/ATO samples, and one commercial Ir nanoparticle. The mass-based activities at 1.8 V were 113 mA/mg for Ir ND1, 116 mA/mg for Ir ND2, and 106 mA/mg for Ir ND3. All performed slightly lower activity compared to the commercial Ir black (154 mA/mg). Since the catalytic activity of Ir NDs followed the order Ir ND2 > Ir ND1 > IrND3, the ATO-supported Ir NDs were only prepared for Ir ND1 and Ir ND2. The mass activities of ATO-supported Ir NDs at 1.8 V were 414 mA/mg for Ir ND1/ATO and 798 mA/mg for Ir ND2/ATO, significantly higher than unsupported Ir NDs. These excellent activities were attributed to the uniformly distributed intrinsically Ir nano-dendritic structure on the surface of ATO nanoparticles. The electrical conductivity of the ATO support was 0.026 ± 0.001 S/cm by the four-point probe method. The high electrical conductivity of the ATO support was expected to be another reason that the ATO-supported Ir NDs had good activity. The corresponding OER polarization curves (normalized to the geometric disk area) of the samples are shown in Figure 7b, and the optimal activities at 1.8 V for unsupported and ATO-supported Ir NDs occurred at Ir ND2 (0.0060 A/cm^2^) and Ir ND2/ATO (0.0106 A/cm^2^), respectively. The Ir-mass-based activity at 1.5, 1.65, and 1.8 V is also displayed in Figure 7c,d for comparison.

A comparison of the mass-based OER activities at 1.8 V on the polarization curves for the 1st and the 100th cycles of LSV is shown in Figure 8. The activity decay of the electrocatalysts increased in the order Ir ND2/ATO < Ir ND1/ATO < Ir ND3 < Ir ND1 < Ir black (Premetek) < Ir ND2. It is evident that the ATO-supported Ir NDs exhibited higher durability than Ir black and unsupported Ir NDs. Specifically, the activity decay of Ir ND2/ATO was only 0.8% after 100 cycles of LSV. The excellent durability of the electrocatalytic activity for Ir ND/ATO was primarily attributed to the in situ synthesis of Ir NDs on the surface of ATO nanoparticles, the uniform dispersion of Ir NDs on the surface of ATO, and good corrosion resistance of ATO in an acidic environment. It is anticipated that the metallic Ir would convert to hydrous Ir oxides upon potential cycling. In addition, strong metal–support interactions would change the electrochemical behavior of the hydrous Ir oxides [10].

The EIS was used to investigate the internal resistances of the electrocatalysts, where the resistance was determined from the width of the semicircles in the *X*-axis. The Nyquist plots obtained from EIS data are shown in Figure 9. Before the LSV measurements, the Nyquist diagrams of Ir NDs only showed diffusion resistance in the electrolyte, about 20 ohm (Figure 9a–c). The charge transfer resistance occurring at the catalyst–electrolyte interface was observed after 100 cycles of LSV, and the resistances of Ir ND1, Ir ND2, and Ir ND3 were approximately 193, 152, and 151 ohm, respectively. This was highly associated with the OER activity decay.

The ATO-supported Ir NDs displayed a different pattern. Except for the diffusion resistance, another semicircle occurred, indicating the resistance of ATO support, which was about 130 ohm for Ir ND1/ATO and 40 ohm Ir ND2/ATO. It is expected that the effect of ATO on the resistance depends on the interactions between Ir NDs and ATO. The interactions were stronger in Ir ND2/ATO; hence, the resistance from the ATO support was less significant. After 100 cycles of LSV, the resistances of the samples increased to about 170 and 43 ohm for Ir ND1/ATO and Ir ND2/ATO, respectively, which was also related to the activity decay.

## 4. Conclusions

The results showed that Ir NDs nanostructures were successfully prepared alone or supported on ATO nanoparticles, where the structures of nanodendrites were dependent on the Ir concentrations in the precursor solution. The XRD patterns of Ir NDs exhibited major diffraction peaks, which were indexed to the reflections of the fcc Ir metal. However, for Ir ND/ATO, the major diffraction peaks of metal Ir were insignificant, indicating the formation of IrOx species on ATO. The unsupported and ATO-supported Ir NDs consisted of metallic Ir as well as Ir oxides, which was supported by the results of XPS. Moreover, the oxidation of ATO stimulated the reduction in Ir during the synthesis processes. The mass-based OER catalytic activity at a potential of 1.8 V had the order Ir ND2/ATO (798 mA/mg) > Ir ND1/ATO (414 mA/mg) > Ir black (155 mA/mg) > Ir ND2 (116 mA/mg) > Ir ND1 (113 mA/mg) > Ir ND3 (106 mA/mg). After performing LSV of 100 cycles, the OER catalytic activity of Ir ND2/ATO remained at > 99% of activity, while Ir ND2 had only about 61.4% of the activity left, indicating that Ir NDs supported on ATO significantly improve the durability of the OER catalytic activity. This observation was also supported by the results of EIS, according to which the charge transfer resistance was the lowest on Ir ND2/ATO.

## Figures and Tables

**Figure 1 nanomaterials-13-02264-f001:**
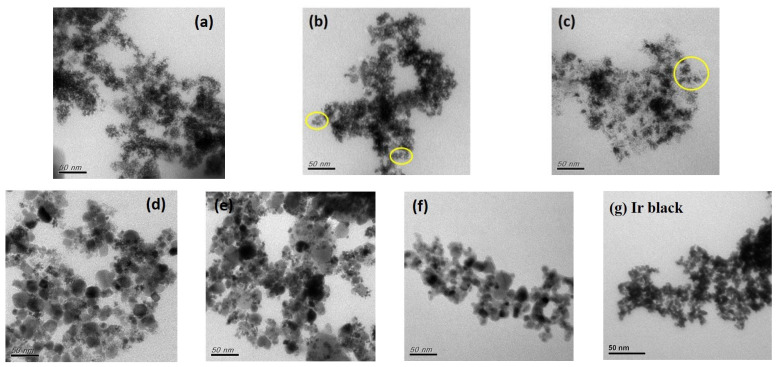
TEM images of the samples: (**a**) Ir ND1; (**b**) Ir ND2; (**c**) Ir ND3; (**d**) Ir ND1/ATO; (**e**) Ir ND2/ATO; (**f**) ATO; (**g**) Ir black (Premetek). The yellow circles show the number or width of branches increased as the concentration of Ir precursor increased.

**Figure 2 nanomaterials-13-02264-f002:**
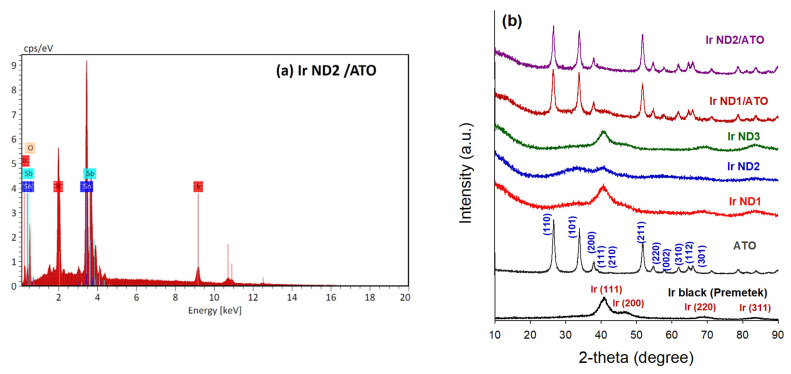
(**a**) EDX pattern of Ir ND2/ATO. (**b**) XRD patterns of all samples.

**Figure 3 nanomaterials-13-02264-f003:**
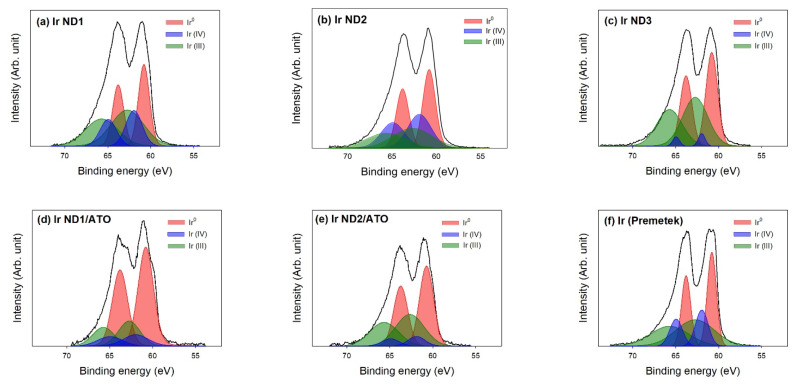
Curve fitting of high-resolution XPS Ir 4f spectra for the samples: (**a**) Ir ND1; (**b**) Ir ND2; (**c**) Ir ND3; (**d**) Ir ND1/ATO; (**e**) Ir ND2/ATO; (**f**) Ir black (Premetek).

**Figure 4 nanomaterials-13-02264-f004:**
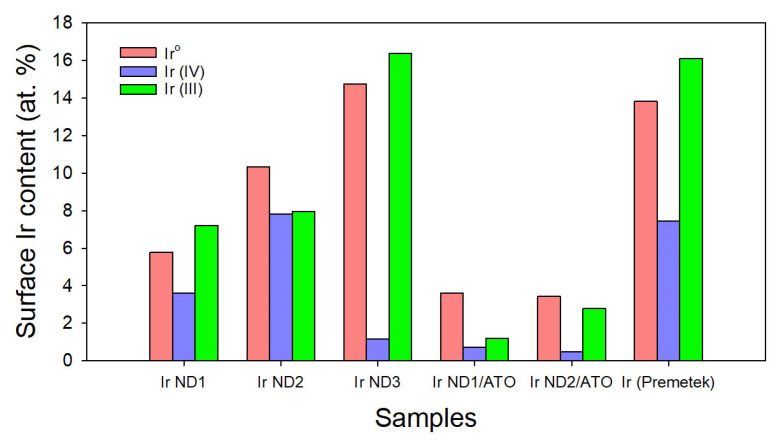
The different chemical states of surface Ir contents from XPS analysis for all samples.

**Figure 5 nanomaterials-13-02264-f005:**
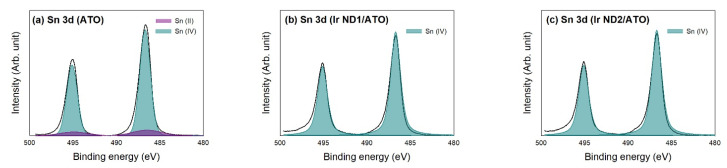
Curve fitting of high-resolution XPS Sn 3d spectra for the samples: (**a**) ATO; (**b**) Ir ND1/ATO; (**c**) Ir ND2/ATO.

**Figure 6 nanomaterials-13-02264-f006:**
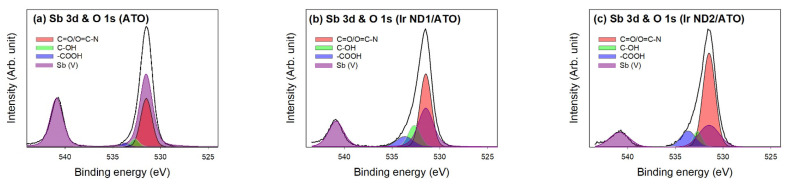
Curve fitting of high-resolution XPS Sb 3d and O 1s spectra for the samples: (**a**) ATO; (**b**) Ir ND1/ATO; (**c**) Ir ND2/ATO.

**Figure 7 nanomaterials-13-02264-f007:**
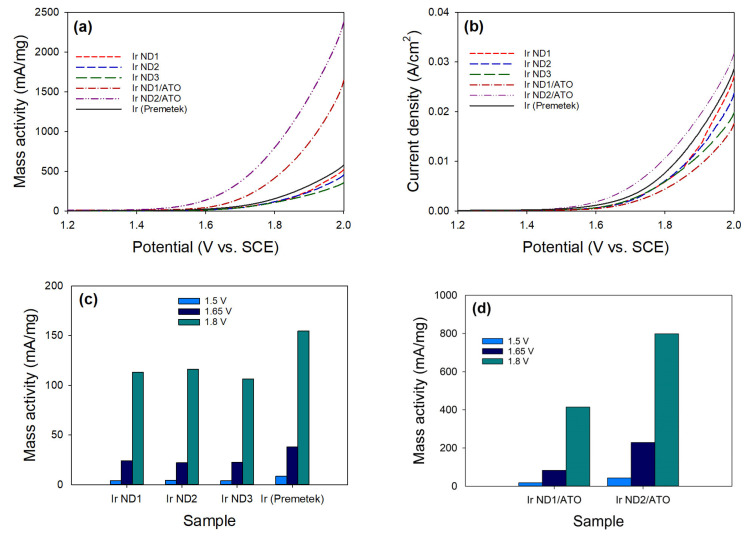
Electrocatalytic OER activity of the catalysts: (**a**) mass activity; (**b**) current density; the Ir-mass-based activity at 1.5, 1.65, and 1.8 V: (**c**) unsupported Ir; (**d**) Ir ND/ATO.

**Figure 8 nanomaterials-13-02264-f008:**
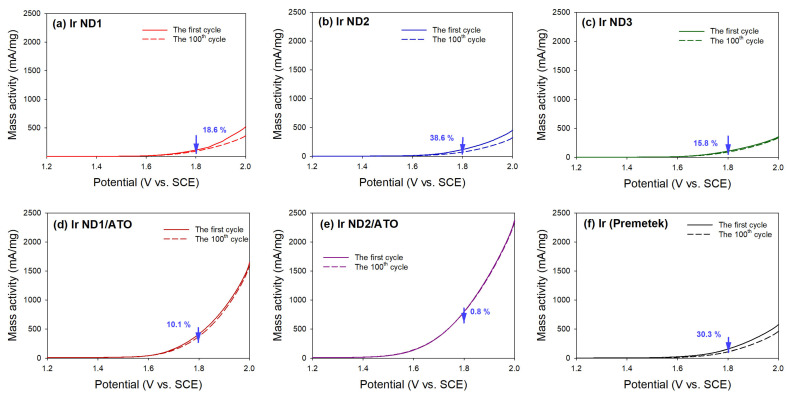
Degradation of electrocatalytic OER activity of the catalysts at 1.8 V: (**a**) Ir ND1; (**b**) Ir ND2; (**c**) Ir ND3; (**d**) Ir ND1/ATO; (**e**) Ir ND2/ATO; (**f**) Ir black (Premetek).

**Figure 9 nanomaterials-13-02264-f009:**
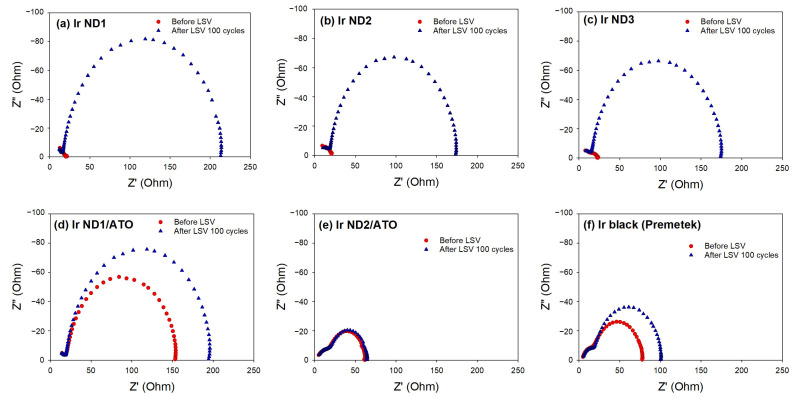
Nyquist plots of the catalysts: (**a**) Ir ND1; (**b**) Ir ND2; (**c**) Ir ND3; (**d**) Ir ND1/ATO; (**e**) Ir ND2/ATO; (**f**) Ir black (Premetek).

**Table 1 nanomaterials-13-02264-t001:** The elemental compositions of the samples from EDX analysis.

Sample	Elemental Composition							
	Weight Percentage (wt.%)				Atomic Percentage (at.%)			
	Ir	O	Sn	Sb	Ir	O	Sn	Sb
Ir ND1	88.8	11.2	—	—	39.8	60.2	—	—
Ir ND2	88.1	11.9	—	—	38.2	61.8	—	—
Ir ND3	95.5	4.5	—	—	63.63	36.37	—	—
Ir ND1/ATO	20.9	13.7	59.5	5.9	7.2	56.5	33.1	3.2
Ir ND2/ATO	26.1	14.6	56.4	2.9	8.7	59.0	30.7	1.6
Ir black (Premetek)	92.8	7.2	—	—	51.7	48.3	—	—
ATO	—	31.3	58.3	10.4	—	77.2	19.4	3.4

**Table 2 nanomaterials-13-02264-t002:** Results of fitting of the XPS Ir 4f region for the samples; values given in at.% of total intensity.

Sample	Ir 4f	Binding Energy (eV)					
		Ir^o^		Ir (IV)		Ir (III)	
		60.8	63.8	61.9	64.9	62.7	65.7
Ir ND1	16.6	3.30	2.48	2.07	1.55	4.11	3.09
Ir ND2	26.1	5.90	4.43	4.47	3.35	4.54	3.41
Ir ND3	32.3	8.43	6.32	0.66	0.50	9.37	7.03
Ir ND1/ATO	5.51	2.07	1.55	0.40	0.30	0.68	0.51
Ir ND2/ATO	6.68	1.96	1.47	0.28	0.21	1.59	1.19
Ir black (Premetek)	37.4	7.90	5.92	4.26	3.20	9.21	6.91

**Table 3 nanomaterials-13-02264-t003:** Results of fitting of the XPS Sn 3d region for the samples; values given in at.% of total intensity.

Sample	Sn 3d	Binding Energy (eV)			
		Sn (II)		Sn (IV)	
		486.5	494.91	486.7	495.11
ATO	23.3	1.71	1.14	12.27	8.18
Ir ND1/ATO	21.1	—	—	12.66	8.44
Ir ND2/ATO	23.32	—	—	13.99	9.33

**Table 4 nanomaterials-13-02264-t004:** Results of fitting of the XPS Sb 3d and O 1s regions for the samples; values given in at.% of total intensity.

Sample	Sb3d	O1s	Binding Energy (eV)				
			Sb (V)		C=O	C-OH	-COOH
			531.5	540.88	531.5	532.7	533.7
ATO	10.9	65.8	6.54	4.36	56.80	6.38	2.62
Ir ND1/ATO	3	70.39	1.80	1.20	46.71	14.46	9.23
Ir ND2/ATO	2.03	67.97	1.22	0.81	51.69	6.02	10.25

## Data Availability

The data presented in this study are available on request from the corresponding author. The data are not publicly available due to privacy restrictions.

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
