# Peer review of "Study on Oxygen Evolution Reaction of Ir Nanodendrites Supported on Antimony Tin Oxide"

_nanomaterials, 2023, doi:10.3390/nano13152264_

Round 1

Reviewer 1 Report

The authors present a study about Ir nanodendrites supported on ATO for OER.

The paper needs major revisions to be published on nanomaterials:

1 - There are several phrases that need to be re-written in a proper, more understandable way. For instance: page 2, line 51. The phrase is not clear and does not mean anything.

2 - Page 2, line 86: "was oxidized to Mo(VI) with suppressing...". Also here the meaning is difficult, if not impossible to understand.

3 - The introduction is too long. Since the paper describes the use of nanodendrites it would be better to focus the introduction mainly on their use and preparation.

4 - Page 6, line 234: " Moreover Ir(III) was though less than..." again, the phrase is not clear.

5 - The whole xps section has to be profoundly revised since the reported fitting are completely wrong and not compatible with what already reported in the literature. Se for instance: ACS Appl. Nano Mater. 2020, 3, 3, 2185–2196.

6 - I don't understand what is the utility of fitting the Sb 3d signal that as the author say is overlapping with the O 1s signal. The results of this analysis is not used in the discussion of the electrocatalytic performances.

7 - The shape of the LSV curves for the unsupported Ir dendrites is very strange and different from what is normally seen in the literature. The stability is also very modest.

8 - The authors should also report long stability tests acquired as chronoamperometry measurements.

The paper needs a strong English editing.

Reviewer 2 Report

The manuscript reports on the synthesis of Ir nanodendrites with and without antimony tin oxide (ATO) support and on the study of their structural, morphological, spectroscopic, electrochemical and impedance spectroscopy properties. The objective is to use the ATO to support Ir in order to build a catalyst that promotes an efficient oxygen evolution reaction for hydrogen production. The manuscript has original results, but needs revisions. I have the following comments:
- On page 3 it is written “one commercial ATO (< 50 nm, 47 m2/g, Aldrich) was dispersed in deionized water (25 ml)”. Was the used ATO a powder ? A plate ? Please, clarify.
- On page 4 it is written “The thickness of the tablets (t, cm) was determined using a precise vernier caliper”. What was the resolution of the scale ?
- On page 4 it is written “When the concentration of Ir precursor increased, the number or the width of branches increased (Figure 1b-c)”. It would be better to put, e.g, arrows in places where this is happening in the figures 1b-c. As it is, it is hard to see the difference.
- On page 5 it is written “For Ir NDs, the Ir content ranged from 88.1 to 95.5 wt.%”. Maybe it would be better to calculate the atomic percentage, since Ir, O and so on, in the samples, have very different molar masses.
- From XRD the authors calculated the grain sizes in the samples, which were in the range 2.2-3.4 nm. How do these values compare with the TEM results in figure 1 ?
- On page 11 it is written “The EIS was used to investigate the internal resistances of the electrocatalysts.” . How was the resistance determined from the EIS results ? From the width of the semicircles in the X-axis ? The authors should clarify this.

The English is good, but has some minor mistakes that would be better to revise.
